# Synthesis and Density Functional Theory Studies of Azirinyl and Oxiranyl Functionalized Isoindigo and (3*Z*,3’*Z*)-3,3’-(ethane-1,2-diylidene)bis(indolin-2-one) Derivatives

**DOI:** 10.3390/molecules24203649

**Published:** 2019-10-10

**Authors:** Gholamhossein Khalili, Patrick M. McCosker, Timothy Clark, Paul A. Keller

**Affiliations:** 1School of Chemistry and Molecular Biosciences, Molecular Horizons, Illawarra Health and Medical Research Institute University of Wollongong, Wollongong, New South Wales 2522, Australia; pmm938@uowmail.edu.au; 2Chemistry Department, Bushehr Branch, Islamic Azad University, PO Box 7519619555 Bushehr, Iran; 3Department of Chemistry and Pharmacy, Computer-Chemistry-Center (CCC), Friedrich-Alexander-Universität Erlangen-Nürnberg (FAU), Nägelsbachstrasse 25, 91052 Erlangen, Germany; Tim.Clark@fau.de

**Keywords:** isoindigo alkylation, organic semiconductor, donor–acceptor functionalization, density functional theory, time dependent density functional theory

## Abstract

The design and synthesis of functionalized isoindigo compounds by reaction of isoindigo with (*S*)-glycidyl tosylate, epibromohydrin, 2-(bromomethyl)-1-(arylsulfonyl)aziridine, and 2-(bromomethyl)-1-(alkylsulfonyl)aziridine in the presence of MeONa proceed under mild conditions in moderate yields. (3*Z*,3’*Z*)-3,3’-(Ethane-1,2-diylidene)bis(1-(oxiran-2-ylmethyl)indolin-2-one), with an extended central olefin π-conjugated moiety was also reacted with methyl-oxiranes to give the corresponding *N,N’*-disubstituted derivative. Calculations with DFT and TD-DFT of hypothetical isoindigo-thiophene DA molecules with various electron withdrawing substituents, including aziridine, oxirane, nitrile, carbonyl, and sulfonate, indicated that the proximity and strength of the functional group have a significant effect on the HOMO, LUMO, vertical excitation energy, and oscillator strength of the π–π* transitions.

## 1. Introduction

The synthesis of new dyes with intense absorption in the visible spectrum is of interest in the development of high-performance organic electronic materials [1,2,3]. For example, donor–acceptor (DA) type semiconductors have been investigated widely for potential application in the design and application of organic-based solar-cell devices. Ideal acceptor building block molecules would possess strong electron withdrawing character, should be solution processable, be of low cost, easily accessible and have excellent stability. 

Isoindigo, the structural isomer of the better-known indigo, has been the subject of numerous studies of solar-cell technologies [4,5,6,7,8,9,10,11,12]. It is an electron-deficient moiety, as indicated by low highest occupied molecular orbital (HOMO) and lowest unoccupied molecular orbital (LUMO) energies, essential for small bandgap DA systems [10]. Furthermore, isoindigo features NH functionalities, which are easily alkylated, allowing facile derivative synthesis [13,14]. This method has been employed to modify a variety of DA systems that have been noted to influence solubility, aggregation, light absorption, blend morphology, charge-carrier mobility, stability, and electronic properties [11,12,15,16,17,18,19,20,21,22,23,24]. 

In the design of DA-semiconductors, *N-*alkylation is primarily used to influence solubility; this has previously been investigated both experimentally and theoretically for isoindigo-based materials [15,16,19,24,25]. These studies, however, did not investigate the effect of functional groups, neglecting their potential influence on the electronics of the DA system. Such effects have been explored in other polymeric systems, and were found primarily to affect the donor and acceptor strength, in addition to film morphology. Extensive experimental screening of new DA-semiconducting molecules can be an expensive process consuming time and resources hence the application of computational theory has become common in screening for new molecules [5,19,25]. Density functional theory (DFT) and the respective time-dependent (TD)-DFT are the most popularly applied theories due to their reasonable accuracy with an attractive computational cost. Within the TD-DFT sphere, the hybrid functional B3LYP combined with Pople basis sets are routinely applied and have generally provided good prediction of singlet excitations [26,27]. However, it has been noted hybrid functionals perform poorly with charge transfer states, which has previously been attributed to isoindigo [28]. To account for this, the long range corrected CAM_B3LYP functional was developed, which has since been considered a more robust method for investigating excited states through TD-DFT, particularly with respect to charge transfer states and oscillator strengths [26,27,28,29,30].

In addition to the calculated vertical excitation energies and their respective oscillator strengths, TD-DFT calculation provides information regarding the molecular orbitals (MOs) involved in the transition. This information can be utilized for qualitative analysis, such as identification of charge transfer states. For the first excitation, the MOs are generally sufficient since the transition is generally only composed of a direct HOMO to LUMO excitation. For higher energy transitions however, the contributions of multiple MOs to an excitation can make their subsequent analysis difficult. To overcome this problem, the natural transition orbital (NTO) transformation was developed, which offers a simplified orbital representation of the electronic transition density matrix by its mathematical diagonalization [31]. Therefore, by transforming each TD-DFT transition density matrix, each vertical excitation can be satisfactorily described by one or two pairs of excited “particle” and empty “hole” orbitals.

We report here the continued exploration of the N-alkylation of isoindigo, with a focus on the incorporation of aziridines and epoxides, which are valuable synthetic building blocks for subsequent synthetic manipulation [32,33,34]. The N-alkylation of the related (3*Z*,3’*Z*)-3,3’-(ethane-1,2-diylidene)bis(indolin-2-one) as an extended π-conjugated dimeric heterocycle was also investigated. Selected isoindigo-aziridine and -oxirane derivatives were then studied using DFT and TD DFT for isoindigo-thiophene DA monomers and dimers and compared with the previously characterized PTI1 [25], alkylated analogue in order to establish the effect of functional groups on photophysical properties. Results were further probed by investigating other proposed *N*-alkylations with electron-withdrawing functional groups, which focused on changes to the calculated vertical excitations.

## 2. Results

### 2.1. Chemical Synthesis and Characterisation

The N-substituted reactions of isoindigo (Scheme 1) and (3*Z*,3’*Z*)-3,3’-(ethane-1,2-diylidene)bis(indolin-2-one) (Scheme 2) with epoxides of (*S*)-glycidyl tosylate, epibromohydrin and aziridine heterocycles such as 2-(bromomethyl)-1-(arylsulfonyl)aziridine, and 2-(bromomethyl)-1-(alkylsulfonyl)aziridine were investigated, using sodium methoxide as a base. Therefore, reaction of isoindigo **1** with (*S*)-glycidyl tosylate or epibromohydrin **2** gave the (*E*)-1,1’-bis(oxiran-2-ylmethyl)-[3,3’-biindolinylidene]-2,2’-diones **3** in 71% and 65% yield respectively.

In a typical example, analysis of the ^13^C NMR spectra of compound **3** showed a resonance at δ 44.1 assigned to C2’’ of epoxide ring. Analysis of the ^1^H NMR spectrum showed two doublets of doublets at δ 3.61 (*J* = 15.1, 5.5 Hz), and δ 4.20 (*J* = 15.0, 3.2 Hz) assigned to H3’’ with protons of CH_2_O assigned to a multiplet at δ 2.60–2.62 and a triplet at δ 2.76 due to the a stereogenic C2’’. Broad absorption bands with maxima at 204 nm and 390 nm were observed in the UV-vis spectrum.

The isoindigo analogue (3*Z*,3’*Z*)-3,3’-(ethane-1,2-diylidene)bis(indolin-2-one) **4** is an extended π-conjugated dimeric heterocycle that has recently been investigated for applications in performance organic semiconductors [35]. Derivatives of this molecule were synthesized by reaction of **4** with (*S*)-glycidyl tosylate or epibromohydrin **2** in the presence of MeONa in DMF at 40 °C for 4 h, which gave (3*Z*,3’*Z*)-3,3’-(ethane-1,2-diylidene)bis(1-(oxiran-2-ylmethyl)indolin-2-one)s **5** in 68% and 62% yield respectively (Table 1). Analysis of the FTIR spectrum of **5** showed an absorption at 1689 cm^−1^ assigned to the two identical carbonyl moieties and the UV-vis spectrum showed two absorption maxima at 201 and 302 nm for this red compound. The HRESI mass spectrum showed a peak at *m/z* 423.1313, assigned to the molecular formula C_24_H_20_N_2_O_4_Na and was indicative of the addition of two 2-methyloxirane units. 

The diverse array of 2-(bromomethyl)-1-(arylsulfonyl)aziridines, and 2-(bromomethyl)-1-(alkylsulfonyl)aziridines **6** constitute a unique subclass of *N*-activated aziridines due to the presence of three electrophilic sites; the exocyclic methylene group and the two carbon atoms of the aziridine moiety [36,37,38]. These valuable precursors, with extensive applications in the synthesis of organic or biological compounds such as *β*-lactams, amino acids, alkaloids, and antibiotics, were synthesized in high yields in two steps starting with allylsulfonamides [39].

The synthesis of the functionalized red-coloured (*E*)-1,1’-bis((1-(alkylsulfonyl)aziridin-2-yl)methyl)-[3,3’-biindolinylidene]-2,2’-diones and (*E*)-1,1’-bis((1-(arylsulfonyl)aziridin-2-yl)methyl)-[3,3’-biindolinylidene]-2,2’-diones **7a–7h** from 2-(bromomethyl)-1-(arylsulfonyl)aziridine, and 2-(bromomethyl)-1-(alkylsulfonyl)aziridine **6** proceeded in 53%–60% yield (Table 1). Analysis of the FTIR spectrum of **7a** showed stretches at 1678 cm^−1^ assigned to the two carbonyl groups while the stretches at 1290 and 1196 cm^−1^ were assigned to the sulfonamide moiety. Analysis of the ^1^H NMR spectrum showed doublets of doublets at δ 3.81 (*J* = 15.0, 5.4 Hz) and 4.08 (*J* = 14.9, 2.4 Hz), doublets at δ 2.20 (*J* = 4.2 Hz) and 2.57 (*J* = 6.9 Hz), a triplet at δ 0.96 ppm (*J* = 7.4 Hz), and three multiplet resonances at δ 1.70–1.79, 2.87–2.91, and 2.99–3.02 assigned to the aliphatic protons of (*E*)-1,1’-bis((1-(propylsulfonyl)aziridin-2-yl)methyl)-[3,3’-biindolinylidene]-2,2’-dione **7a**. The ^13^C NMR spectrum showed two resonances at δ 35.11 and 35.13 assigned to C3’’ and resonances at δ 39.43 and 39.45 ppm were assigned to C1’’ indicating a mixture of diastereomers present due to the stereogenicity of aziridine C2’’. The UV-vis spectrum showed two absorption maxima at 203 and 390 nm for this red compound.

The utility of the protocol was extended to the reaction of isoindigo **1** with (1*S*,4*S*/*R*)-1-(((2-(bromomethyl)aziridin-1-yl)sulfonyl)methyl)-7,7-dimethylbicyclo[2.2.1]heptan-2-one **6i** produced from enantiomerically pure (*S*)-10-camphor sulfonyl chloride as mixtures of *SR* and *SS* isomers in DMF at 35–40 °C to give the (*E*)-1,1’-bis((1-((((1*S*,4S/*R*)-7,7-dimethyl-2-oxobicyclo[2.2.1]heptan-1-yl)methyl)sulfonyl)aziridin-2-yl)methyl)-[3,3’-biindolinylidene]-2,2’-dione **7i** in 52% yield (Table 1, entry 9). Analysis of the FTIR spectrum of **7i** showed absorptions at 1740 and 1693 cm^−1^ assigned to the carbonyls of the ketone and amide moieties. Analysis of the ^13^C NMR spectrum revealed four resonances at 166.90, 166.94, 212.6, and 213.1 ppm assigned to carbonyls of the amide and ketone respectively. These observations are in agreement with the formation of the stable 7**i** as two diastereomers. The presence of two singlet resonances at δ 0.69 and 0.97 with an integration equivalent to two methyl groups in ^1^H NMR spectrum is a further confirmation of the synthesis of the *SR* and *SS* isomers. The HRMS spectrum of this molecule displayed a peak at 799.2875, assigned to molecular formula C_42_H_48_N_4_O_8_S_2_.

### 2.2. Computational Studies

To investigate the theoretical application of the synthesized *N-*alkylated molecules in DA-molecules, molecular modelling of theoretical thiophene isoindigo molecules was performed on DFT optimized monomer and dimer structures (Figure 1). Only the substitution of mesyl protected aziridine (Me-aziridine) and oxirane (Me-oxirane) were modelled, and these were compared with an *iso*-butyl substitution (PTI1), which has previously been characterized [25].

Analysis of the TD-DFT calculations indicated that substitution with the Me-oxirane had negligible effect on the vertical excitations and therefore the modelled UV-vis absorption (Figure 2). The aziridine substitution did, however, result in a red shift and increased oscillator strength for the first vertical excitation. Despite the methyl group separating the aziridine ring from the isoindigo unit, it is apparent that functional groups do influence the electronics of the principle DA system.

To probe and obtain a better understanding of this *N*-substitution effect, a series of electron-withdrawing substitutions were subsequently modelled, including nitrile, carbonyl, and sulfonate-based substitutions (Figure 1). Comparison of the optimized monomer geometries revealed substitutions with a greater electron withdrawing character altered C3-C3’ and C2’’-C6 torsion angles and resulted in a more twisted core (Figure 3). The Me-aziridine substitution was the exception to this trend, wherein the change was towards a more planar geometry. This is not likely to be a steric effect, since the *N*-substitution groups are well placed facing out from the system. Nevertheless, because of the separation of the aziridine moiety from the DA system core, this flatter geometry is likely the reason for the observed energy red shift and oscillator strength increase for the first vertical transition.

Comparison of the modelled absorption spectra revealed a general energy red shift and oscillator strength increase for the first vertical excitation, with a decrease in oscillator strength for the next prominent vertical excitation. The most prominent difference between the monomer and dimer vertical excitations was a general red shift of the energies, in addition to an almost 3-fold increase in the oscillator strength of the first vertical excitation, which was interpreted as being due to increasing the size of the π system, in agreement with theory and experiment [5].

The changing oscillator strengths was a distinct observation. It has been previously noted that increasing the number of electrons in the molecule, in our case with the substituents, will give an increase of oscillator strength in TD-DFT calculations [40]. However, it was observed here that only specific transitions underwent significant change, and did not always increase in oscillator strength. Furthermore, molecular modelling of thiophene-isoindigo monomers with hexyl carbonyl N-substitutions, where the position of the carbonyl is changed, indicated the calculated vertical excitations are directly influenced by the proximity of the carbonyls (Figure 4).

As oscillator strength is related to the transition dipole moment [30], it is possible the MOs involved are changing such that the orbital overlap between the ground and excited state are different. Therefore, to investigate this proposition further, the character of these transitions, the MOs and NTOs were calculated for *iso*-butyl, acetonitrile and mesylate substituted molecules, as a sample of the complete set of molecules studied. In all TD-DFT calculations, the 1st vertical excitation was found to consist primarily of the HOMO and LUMO, which were found to be π/π* in character, in agreement with previous studies (Figure 5 and Appendix A) [4,5,6]. Visual comparison of these molecular orbitals indicated similar shape and distribution, with delocalization of π and π* orbitals on the isoindigo and thiophene moieties.

The MO analysis indicated the principle MOs are consistent, independent of the specific *N*,*N*´-substitutive groups present, and therefore the primary changes observed can be related to the inductive influences of the respective substitutions. Inspection of the calculated orbital energy levels for the monomers indicated similar stabilization of the HOMO and LUMO for weaker inductive groups, i.e., Me-oxirane or Me-aziridine, up to 0.05 eV. Derivatives with stronger inductive groups, i.e., the sulfonates and carbonyls, showed greater LUMO stabilization, up to 0.36 eV (Appendix A).

To analyze the next major vertical excitation, NTO analysis was used to simplify the mixed occupied component. This π/π* transition was similar to the 1st excitation, however, minimal electron density is present on the thiophene moiety, and disappears completely with mesylate substitution (Figure 6). This is particularly interesting in relation to the oscillator strengths calculated, whereby the 1st vertical excitation increased, this excitation has decreased. Relating this back to the observed changes in geometry, it can be rationalized in terms of conjugation, whereby the 1st excitation is delocalized over both isoindigo and thiophene, the increased planarity of the thiophene-isoindigo bond improves the orbital overlap and conjugation. The 3rd excitation however, is localized on the isoindigo because twisting has hindered orbital overlap and conjugation. This can be seen in the NTO analysis, where the π orbital seen in the *iso*-butyl substituted isoindigo central C2=C2´ alkene loses orbital overlap with acetonitrile or mesylate substitution and resulted in more *p*-type orbitals over the C2 and C2´carbons.

From the MO and NTO analyses performed here no significant patterns were found to relate the electron density distribution and the TD-DFT patterns observed. Like the 1st excited state HOMO to LUMO transition, substitutions with electron-withdrawing groups lower the energy of the principle π-states. Based on the TD-DFT excitation patterns, in agreement with the changes observed in the HOMO and LUMO energy levels, the result reported here is in agreement with those previously found for substitutive effects in other molecules [5,18]. That is, *N*-alkylation of isoindigo in DA-semiconductor molecules has an opto-electronic stabilizing effect on the MOs, particularly the LUMO when electron-withdrawing groups are used, however they are not directly involved in any transitions. Therefore, the *N*-alkylation of isoindigo with electron-withdrawing substituents red-shifts the first vertical optical transition and could provide useful chromophores for subsequent reactions or functionalization.

## 3. Conclusions

An efficient synthesis of the functionalized red-colored (*E*)-1,1’-bis((1-(alkylsulfonyl)aziridin-2-yl)methyl)-[3,3’-biindolinylidene]-2,2’-diones **7a–d** and (*E*)-1,1’-bis((1-(arylsulfonyl)aziridin-2-yl)methyl)-[3,3’-biindolinylidene]-2,2’-diones **7e–h** has been accomplished starting from accessible *N*-activated Me-aziridine substrates using inexpensive sodium methoxide as base at room temperature. This approach was applied successfully to the synthesis of (*E*)-1,1’-bis(oxiran-2-ylmethyl)-[3,3’-biindolinylidene]-2,2’-dione by the incorporation of the appropriate leaving group, such as Br or OTs, into the epoxide substrate.

To investigate the applicability of these molecules in DA based solar cells, molecular modelling showed these *N*-substitutions had minimal influence on the electronics of the primary DA system. Further computational investigations indicated that the proximity of functional groups with the DA system is important, and proposed acetonitrile, carbonyl or sulfonate substitutions can have a substantial effect. Briefly, this included a lowering of the HOMO and particularly the LUMO energies, with a respective increase in oscillator strength and decrease in energy for the 1st vertical excitation (HOMO to LUMO).

## 4. Experimental Section

### 4.1. General Methods

Reagents and solvents were purchased reagent grade and used without further purification except 2-(bromomethyl)-1-(arylsulfonyl)aziridine and 2-(bromomethyl)-1-(alkylsulfonyl)aziridine (1.5 mmol), 2-(bromomethyl)-1-tosylaziridine, 2-(bromomethyl)-1-(phenylsulfonyl)aziridine, 2-(bromomethyl)-1-(methylsulfonyl)aziridine. All reactions were performed in standard oven dried glassware under a nitrogen atmosphere unless otherwise stated. Melting point temperatures are expressed in degrees Celsius (°C) and are uncorrected. ^1^H and ^13^C NMR spectra (CDCl_3_) were recorded at 400 MHz and 100.6 MHz respectively with chemical shifts (δ) reported in parts per million relative to TMS (δ = 0 ppm) or CDCl_3_ (δ = 77.0 ppm) as internal standards. Coupling constants (*J*) are reported in Hertz (Hz). Multiplicities are reported as singlet (s), broad singlet (bs), doublet of doublets (dd), or multiplet (m). High Resolution Electrospray Ionization (HRESI-single quadrupole) mass spectra have their ion mass to charge (*m*/*z*) values stated with their relative abundances as a percentage in parentheses. Peaks assigned to the molecular ion are denoted by [M + H]^+^ or [M + Na]^+^. Infrared (IR) spectra were recorded on neat samples. UV-visible spectra were recorded CH_2_Cl_2_ solutions. Thin Layer Chromatography (TLC) was performed using Silica Gel F_254_ aluminum sheets. Column chromatography was performed under gravity using Silica Gel 60 (0.063–0.200 mm). Eluents are in volume to volume (*v*:*v*) proportions. Solvent extracts or chromatographic fractions were concentrated by rotary evaporation *in vacuo*.

### 4.2. Synthesis and Characterisation

*(E)-1,1’-bis(oxiran-2-ylmethyl)-[3,3’-biindolinylidene]-2,2’-dione (***3***)*. To a stirred solution of isoindigo 1 (131 mg, 0.5 mmol) in DMF (3 mL) was added sodium methoxide solution (59.4 mg, 1.0 mmol) in DMF (2 mL) at room temperature. The reaction mixture stirred for 20 min, followed by the addition of the (*S*)-glycidyltosylate or epibromohydrin (1.5 mmol) and the reaction stirred for 5 h at 40 °C. The mixture was poured onto H_2_O (10 mL), extracted with CH_2_Cl_2_ (20 mL), dried (MgSO_4_), and the solvent was removed under reduced pressure. The residue was subjected to silica gel (0.063–0.200 size) column chromatography using hexane-ethyl acetate as eluent. Red powder, m.p. 158 °C. IR (KBr) (ν*_max_*/cm^−1^): 1682, 1604, 1464, 1353, 846, 774. λ*_max_*/nm (ɛ, M^−1^ cm^−1^) 204 (39120), 390 (1091). ^1^H NMR (400.1 MHz, CDCl_3_): δ 2.60–2.62 (m, 2H), 2.76 (t, *J* = 4.3 Hz, 2H), 3.13–3.15 (m, 2H), 3.61 (dd, *J* = 15.1, 5.5 Hz, 2H), 4.20 (dd, *J* = 15.0, 3.2 Hz, 2H), 6.87 (d, *J* = 7.8 Hz, 2H), 6.98 (t, *J* = 7.6 Hz, 2H), 7.28 (t, *J* = 7.6 Hz, 2H), 9.07 (d, *J* = 7.9 Hz, Hz, 2H). ^13^C NMR (100.6 MHz, CDCl_3_): δ 41.0, 44.1, 48.8, 107.6, 120.4, 121.5, 128.8, 131.7, 132.2, 143.6, 166.9. HRMS (ESI) [M + Na]^+^ calcd for C_22_H_18_N_2_O_4_Na 397.1164, found, 397.1176.

*(3Z,3’Z)-3,3’-(ethane-1,2-diylidene)bis(1-(oxiran-2-ylmethyl)indolin-2-one) (***5***)*. To a stirred solution of (3*Z*,3’*Z*)-3,3’-(ethane-1,2-diylidene)bis(indolin-2-one) 4 (131 mg, 0.5 mmol) in DMF (3 mL) was added sodium methoxide solution (59.4 mg, 1.0 mmol) in DMF (2 mL) at room temperature. The reaction mixture stirred for 20 min, followed by the addition of the (*S*)-glycidyltosylate or epibromohydrin (1.5 mmol) and the reaction stirred for 5 h at 40 °C. The mixture was poured onto H_2_O (10 mL), extracted with CH_2_Cl_2_ (20 mL), dried (MgSO_4_), and the solvent was removed under reduced pressure. The residue was subjected to silica gel (0.063–0.200 size) column chromatography using hexane-ethyl acetate as eluent. Red powder, m.p. 150 °C. IR (KBr) (ν*_max_*): 1689, 1603, 1465, 1346, 1093, 711, 679 cm^-1^. λ*_max_*/nm (ɛ, M^−1^ cm^−1^) 201 (2767), 302 (2929). ^1^H NMR (400.1 MHz, CDCl_3_): δ 2.61-2.63 (m, 2H), 2.78 (t, *J* = 4.1 Hz, 2H), 3.16 (s, 2H), 3.60 (dd, *J* = 15.0, 5.4 Hz, 2H), 4.18 (dd, *J* = 15.1, 2.9 Hz, 2H), 6.90 (d, *J* = 7.8 Hz, 2H), 6.98 (t, *J* = 7.5 Hz, 2H), 7.23 (t, *J* = 7.7 Hz, 2H), 7.55 (d, *J* = 7.4 Hz, 2H), 8.91 (s, 2H). ^13^C NMR (100.6 MHz, CDCl_3_): δ 40.6, 44.1, 48.9, 108.1, 120.2, 121.5, 122.0, 127.9, 129.7, 129.8, 141.9, 166.1. HRMS (ESI) [M + Na]^+^ calcd for C_24_H_20_N_2_O_4_Na 423.1321, found, 423.1313.

#### 4.2.1. General procedure for the synthesis of **7**

To a magnetically stirred solution of isoindigo 1 (131 mg, 0.5 mmol) in DMF (3 mL) was added sodium methoxide solution (59.4 mg, 1.0 mmol) in DMF (2 mL) at room temperature. The reaction mixture stirred for 20 min, followed by the addition of the 2-(bromomethyl)-1-(arylsulfonyl)aziridine, or 2-(bromomethyl)-1-(alkylsulfonyl)aziridine (1.5 mmol) and the reaction stirred for 12 h at room temperature. The mixture was poured onto H_2_O (10 mL), extracted with CH_2_Cl_2_ (20 mL), dried (MgSO_4_), and the solvent was removed under reduced pressure. The residue was subjected to silica gel (0.063–0.200 size) column chromatography using hexane-ethyl acetate as eluent.

*(E)-1,1’-bis((1-(propylsulfonyl)aziridin-2-yl)methyl)-[3,3’-biindolinylidene]-2,2’-dione* (**7a**). Red powder, Yield 58%, m.p. 125 °C. IR (KBr) (ν*_max_*): 1678, 1606, 1464, 1315, 1290, 1259, 1230, 1196, 1139, 681 cm^–1^. λ*_max_*/nm (ɛ, M^−1^ cm^−1^) 203 (1262), 390 (416). ^1^H NMR (400.1 MHz, CDCl_3_): δ 0.86 (t, *J* = 7.4 Hz, 6H), 1.70–1.79 (m, 4H), 2.20 (d, *J* = 4.2 Hz, 2H), 2.57 (d, *J* = 6.9 Hz, 2H), 2.87–2.91 (m, 4H), 2.99–3.02 (m, 2H), 3.81 (dd, *J* = 15.0, 5.4 Hz, 2H), 4.08 (dd, *J* = 14.9, 2.4 Hz, 2H), 6.85 (d, *J* = 7.7 Hz, 2H), 7.01 (t, *J* = 7.7 Hz, 2H), 7.31 (t, *J* = 7.5 Hz, 2H), 9.06 (d, *J* = 7.9 Hz, 2H). ^13^C NMR (100.6 MHz, CDCl_3_): δ 11.8, 15.8, 29.9, 35.11, 35.13, 39.43, 39.45, 53.1, 107.4, 120.4, 121.7, 128.9, 131.7, 132.0, 143.2, 166.9. HRMS (ESI) [M + H]^+^ calcd for C_28_H_32_N_4_O_6_S_2_Na 607.1661, found, 607.1652.

*(E)-1,1’-bis((1-(isopropylsulfonyl)aziridin-2-yl)methyl)-[3,3’-biindolinylidene]-2,2’-dione (***7b***).* Red powder, Yield 57%, m.p. 142 °C. IR (KBr) (ν*_max_*/cm^−1^): 1691, 1607, 1466, 1354, 1307, 1260, 1095, 1017, 796. λ*_max_*/nm (ɛ, M^−1^ cm^−1^) 219 (1228), 272 (1324). ^1^H NMR (400.1 MHz, CDCl_3_): δ 1.30 (d, *J* = 5.4 Hz, 12H), 2.20 (d, *J* = 4.2 Hz, 2H), 2.56 (d, *J* = 6.9 Hz, 2H), 3.01–3.02 (m, 2H), 3.12 (q, *J* = 6.8 Hz, 2H), 3.93 (dd, *J* = 15.0, 4.6 Hz, 2H), 4.03 (dd, *J* = 10.6, 3.2 Hz, 2H), 6.85 (d, *J* = 7.8 Hz, 2H), 7.01 (t, *J* = 7.8 Hz, 2H), 7.31 (t, *J* = 7.2 Hz, 2H), 9.06 (d, *J* = 7.9 Hz, 2H). ^13^C NMR (100.6 MHz, CDCl_3_): δ 15.2, 15.3, 30.7, 34.1, 39.2, 52.6, 107.5, 120.4, 121.7, 128.8, 131.6, 132.1, 143.3, 167.0. HRMS (ESI) [M + Na]^+^ calcd for C_28_H_32_N_4_O_6_S_2_Na 607.1661, found, 607.1691.

*(E)-1,1’-bis((1-(ethylsulfonyl)aziridin-2-yl)methyl)-[3,3’-biindolinylidene]-2,2’-dione (*7c*).* Red powder, Yield 59%, m.p. 142 °C. IR (KBr) (ν*_max_*/cm^−1^): 1690, 1607, 1465, 1312, 1142, 741. λ*_max_*/nm (ɛ, M^−1^ cm^−1^) 202 (2972), 273 (1742). ^1^H NMR (400.1 MHz, CDCl_3_): δ 1.34 (t, *J* = 7.4 Hz, 6H), 2.28 (d, *J* = 4.2 Hz, 2H), 2.65 (d, *J* = 6.9 Hz, 2H), 3.01–3.11 (m, 6H), 3.92 (dd, *J* = 14.5, 4.9 Hz, 2H), 4.15 (dd, *J* = 15.0 Hz, 4.0 Hz, 2H), 6.93 (d, *J* = 7.8 Hz, 2H), 7.08 (t, *J* = 7.6 Hz, 2H), 7.39 (t, *J* = 8.0 Hz, 2H), 9.13 (d, *J* = 7.6 Hz, 2H). ^13^C NMR (100.6 MHz, CDCl_3_): δ 7.7, 31.2, 35.96, 35.98, 40.41, 40.44, 47.0, 108.5, 121.4, 122.8, 129.9, 132.7, 133.1, 144.2, 168.0. HRMS (ESI) [M + Na]^+^ calcd for C_26_H_28_N_4_O_6_S_2_Na 579.1348, found, 579.1374.

*(E)-1,1’-bis((1-(methylsulfonyl)aziridin-2-yl)methyl)[3,3’-biindolinylidene]-2,2’-dione (*7d*).* Red powder, Yield 60%, m.p. 171 °C. IR (KBr) (ν*_max_*/cm^−1^): 1680, 1603, 1463, 1141, 1097, 973, 773. λ*_max_*/nm (ɛ, M^−1^ cm^−1^) 203 (1254), 272 (889). ^1^H NMR (400.1 MHz, CDCl_3_): δ 2.32 (d, *J* = 4.2 Hz, 2H), 2.67 (d, *J* = 6.9 Hz, 2H), 2.90 (m, 6H), 3.08 (s, 2H), 3.81–3.86 (m, 2H), 4.20 (dd, *J* = 15.0, 4.0 Hz, 2H), 6.95 (d, *J* = 7.7 Hz, 2H), 7.09 (t, *J* = 7.6 Hz, 2H), 7.40 (t, *J* = 7.5 Hz, 2H), 9.14 (d, *J* = 7.7 Hz, 2H). ^13^C NMR (100.6 MHz, CDCl_3_): δ 31.0, 36.8, 39.6, 40.5, 108.5, 121.4, 121.9, 130.0, 132.8, 133.1, 144.1, 168.0. HRMS (ESI) [M + Na]^+^ calcd for C_26_H_28_N_4_O_6_S_2_Na 551.1035, found, 551.1060.

*(E)-1,1’-bis((1-(m-tolylsulfonyl)aziridin-2-yl)methyl)[3,3’-biindolinylidene]-2,2’-dione (*7e*).* Red powder, Yield 57%, m.p. 138 °C. IR (KBr) (ν*_max_*/cm^−1^): 1682, 1606, 1466, 1354, 1155, 1091, 740, 687. λ*_max_*/nm (ɛ, M^−1^ cm^−1^) 223 (1740), 272 (868). ^1^H NMR (400.1 MHz, CDCl_3_): δ 2.08 (s, 6H), 2.28 (d, *J* = 4.2 Hz, 2H), 2.77–2.81 (m, 2H), 3.03–3.05 (m, 2H), 3.44–3.55 (m, 2H), 4.03–4.41 (m, 2H), 6.62 (d, *J* = 7.8 Hz, 2H), 6.92–7.16 (m, 6H), 7.19 (d, *J* = 6.2 Hz, 2H), 7.16–7.20 (m, 2H), 7.40–7.44 (m, 2H), 8.90 (d, *J* = 7.9 Hz, 2H). ^13^C NMR (100.6 MHz, CDCl3): δ 20.1, 30.2, 36.6, 39.7, 39.9, 107.2, 107.3, 120.2, 121.4, 123.7, 127.0, 127.6, 128.7, 128.8, 131.4, 131.6, 133.5, 135.9, 138.2, 142.7, 166.5. HRMS (ESI) [M + Na]^+^ calcd for C_36_H_32_N_4_O_6_S_2_Na 703.1661, found, 703.1674.

*(E)-1,1’-bis((1-tosylaziridin-2-yl)methyl)[3,3’-biindolinylidene]-2,2’-dione (*7f*).* Red powder, Yield 53%, m.p. 144 °C. IR (KBr) (ν*_max_*/cm^−1^): 1688, 1603, 1466, 1321, 1259, 1175, 1088, 1014, 793. λ*_max_*/nm (ɛ, M^−1^ cm^−1^) 212 (1438), 272 (966). ^1^H NMR (400.1 MHz, CDCl_3_): δ 2.10 (b, 6H), 2.28 (s, 2H), 2.79 (t, *J* = 6.2 Hz, 2H), 3.00 (s, 2H), 3.35–3.47 (m, 2H), 4.08 (td, *J* = 14.3, 2.6 Hz, 2H), 6.60–6.61 (m, 2H), 6.85 (t, *J* = 7.7 Hz, 4H), 6.95 (t, *J* = 7.6 Hz, 2H), 7.19 (b, 2H), 7.47 (t, *J* = 7.16 Hz, 4H), 8.92 (d, *J* = 7.8 Hz, 2H). ^13^C NMR (100.6 MHz, CDCl_3_): δ 20.4, 30.1, 30.2, 36.7, 36.8, 39.9, 40.0, 107.3, 107.4, 120.2, 121.3, 126.5, 128.2, 128.36, 128.39, 131.56, 131.59, 133.0, 133.1, 142.7, 143.7, 166.5 HRMS (ESI) [M + Na]^+^ calcd for C_36_H_32_N_4_O_6_S_2_Na 703.1661, found, 703.1655.

*(E)-1,1’-bis((1-(phenylsulfonyl)aziridin-2-yl)methyl)[3,3’-biindolinylidene]-2,2’-dione (*7g*).* Red powder, Yield 54%, m.p. 148 °C. IR (KBr) (ν*_max_*/cm^−1^): 1679, 1612, 1435, 1353, 1318, 1084, 745. λ*_max_*/nm (ɛ, M^−1^ cm^−1^) 218 (1251), 245 (1857). ^1^H NMR (400.1 MHz, CDCl_3_): δ 2.36–2.37 (m, 2H), 2.84–2.87 (m, 2H), 3.10–3.12 (m, 2H), 3.51–3.60 (m, 2H), 4.08–4.15 (m, 2H), 6.67 (d, *J* = 7.8, 2H), 7.01 (t, *J* = 7.7 Hz, 2H), 7.16–7.21 (m, 6H), 7.23–7.35 (m, 2H), 7.68 (d, *J* = 8.0 Hz, 2H), 8.93 (d, *J* = 7.9 Hz, 2H). ^13^C NMR (100.6 MHz, CDCl_3_): δ 31.40, 31.43, 37.66, 37.69, 40.7, 40.8, 108.30, 108.34, 121.3, 122.4, 127.5, 128.75, 128.77, 129.64, 129.69, 132.5, 132.7, 133.7, 137.0, 137.1, 143.7, 169.5. HRMS (ESI) [M + Na]^+^ calcd for C_36_H_32_N_4_O_6_S_2_Na 675.1345, found, 675.1371.

*(E)-1,1’-bis((1-((4-isopropylphenyl)sulfonyl)aziridin-2-yl)methyl)-3,3’-biindolinylidene]-2,2’-dione (*7h*).* Red powder, Yield 56%, m.p. 136 °C. IR (KBr) (ν*_max_*/cm^−1^): 1691, 1603, 1321, 1158, 1092, 744. λmax/nm (ɛ, M^−1^ cm^−1^) 211 (634), 272 (604). ^1^H NMR (400.1 MHz, CDCl_3_): δ 1.15 (b, 12H), 2.32 (s, 2H), 2.78 (d, *J* = 6.7 Hz, 2H), 3.12 (s, 2H), 3.47–3.48 (m, 2H), 3.63–3.68 (m, 2H), 4.09–4.12 (m, 2H), 6.72 (d, *J* = 7.5 Hz, 2H), 7.02 (t, *J* = 7.3 Hz, 2H), 7.13 (s, 4H), 7.26 (s, 2H), 7.66 (d, *J* = 7.2 Hz, 4H), 8.99 (d, *J* = 7.4 Hz, 2H). ^13^C NMR (100.6 MHz, CDCl_3_): δ 22.3, 22.4, 30.3, 32.9, 36.4, 39.7, 107.4, 120.2, 121.5, 125.9, 126.8, 128.5, 131.5, 131.7, 133.5, 142.9, 154.1, 166.7. HRMS (ESI) [M + H]^+^ calcd for C_40_H_41_N_4_O_6_S_2_ 737.2468, found, 737.2501.

(E)-1,1’-bis((1-((((1S,4S/R)-7,7-dimethyl-2-oxobicyclo[2.2.1]heptan-1-yl)methyl)sulfonyl)aziridin-2-yl)methyl)[3,3’-biindolinylidene]-2,2’-dione (7i). Red powder, Yield 52%, m.p. 186 °C. IR (KBr) (ν_max_/cm^−1^): 1740, 1693, 1607, 1349, 1146, 1097, 665. λ_max_/nm (ɛ, M^−1^ cm^−1^) 201 (2550), 272 (1110). ^1^H NMR (400.1 MHz, CDCl_3_): δ 0.69 (s, 6H), 0.92 (s, 6H), 1.29–1.34 (m, 2H), 1.84–1.99 (m, 8H), 2.25 (b, 5H), 2.60–2.62 (m, 2H), 2.80–2.87 (m, 2H), 3.02–3.09 (m, 2H), 3.38–3.53 (m, 3H), 3.80–4.11 (m, 4H), 6.89 (d, *J* = 7.2 Hz, 2H), 6.98 (d, *J* = 7.2 Hz, 2H), 7.30 (d, *J* = 7.6 Hz, 2H), 9.02–9.05 (m, 2H). ^13^C NMR (100.6 MHz, CDCl_3_): δ 14.2, 18.5, 18.61, 18.69, 18.7, 23.7, 25.8, 30.3, 30.5, 35.5, 35.9, 39.30, 39.36, 39.42, 39.46, 41.3, 41.4, 41.5, 46.9, 47.0, 48.0, 48.2, 57.1, 64.8, 107.4, 107.5, 120.5, 121.6, 121.7, 129.0, 129.1, 131.5, 131.6, 131.7, 132.1, 143.13, 143.17, 143.2, 166.90, 166.94, 212.6, 213.1. HRMS (ESI) [M − H]^+^ calcd for C_42_H_47_N_4_O_8_S_2_ 799.2835, found, 799.2875.

#### 4.2.2. General procedure for the synthesis of **6**

To a magnetically stirred solution of *N*-allylarylsulfonamide or *N*-allylalkylsulfonamide (20 mmol) in dry dichloromethane (25 mL) at 0 °C was added dropwise a solution bromine (25 mmol) in dichloromethane (3 mL). The reaction mixture stirred for 30 min at 0 °C and the solvent was removed in vacuo. The mixture was dissolved in ethanol (15 mL) and added to aqueous NaOH solution (1 M, 100 mL), stirred for 10 min at room temperature and extracted with ether. The organic solution was dried (MgSO_4_), and the solvent removed under reduced pressure to give the products **6**. The spectroscopic data for compounds **6d**, **6g**, and **6e** are in agreement with that reported [41,42].

*2-(bromomethyl)-1-(propylsulfonyl)aziridine (*6a*).* Colorless oil, Yield 75%. IR (KBr) (ν*_max_*/cm^−1^): 1316, 1143, 932, 794. ^1^H NMR (400.1 MHz, CDCl_3_): δ 1.10 (t, *J* = 7.4 Hz, 3H), 1.93–2.00 (m, 2H), 2.28 (d, *J* = 4.2 Hz, 1H), 2.77 (d, *J* = 6.8 Hz, 1H), 3.01–3.04 (m, 1H), 3.16–3.31 (m, 2H), 3.49–3.53 (m, 2H). ^13^C NMR (100.6 MHz, CDCl_3_): δ 12.9, 16.8, 31.4, 33.2, 39.6, 53.8.

*2-(bromomethyl)-1-(isopropylsulfonyl)aziridine (*6b*).* Colorless oil, Yield 80%. IR (KBr) (ν*_max_*/cm^−1^): 1313, 1139, 931, 878, 724, 629. ^1^H NMR (400.1 MHz, CDCl_3_): δ 1.39 (d, *J* = 6.8 Hz, 3H), 1.43 (d, *J* = 6.8 Hz, 3H), 2.20 (d, *J* = 4.4 Hz, 1H), 2.70 (d, *J* = 6.8 Hz, 1H), 2.96–3.00 (m, 1H), 3.27–3.38 (m, 3H). ^13^C NMR (100.6 MHz, CDCl_3_): δ 15.9, 16.6, 31.2, 33.6, 39.0, 53.5.

*2-(bromomethyl)-1-(ethylsulfonyl)aziridine (*6c*)*. Colorless oil, Yield 82%. IR (KBr) (ν*_max_*/cm^−1^): 1316, 1142, 931. ^1^H NMR (400.1 MHz, CDCl_3_): δ 1.42 (t, *J* = 7.6 Hz, 3H), 2.22 (d, *J* = 4.0 Hz, 1H), 2.72 (d, *J* = 6.8 Hz, 1H), 2.96–2.99 (m, 1H), 3.12-3.26 (m, 3H), 3.44 (dd, *J* = 9.2 Hz, 4.8 Hz, 1H). ^13^C NMR (100.6 MHz, CDCl_3_): δ 6.8, 30.4, 32.3, 38.7, 45.8.

*2-(bromomethyl)-1-(m-tolylsulfonyl)aziridine (*6f*).* Colorless oil, Yield 85%. IR (KBr) (ν*_max_*/cm^−1^): 1324, 1305, 1152, 872, 687. ^1^H NMR (400.1 MHz, CDCl_3_): δ 2.25 (d, *J* = 8.4 Hz, 1H), 2.44 (s, 3H), 2.82 (d, *J* = 6.8 Hz, 1H), 3.08–3.14 (m, 1H), 3.27-3.29 (m, 2H), 7.43–7.46 (m, 2H), 7.74–7.78 (m, 2H). ^13^C NMR (100.6 MHz, CDCl_3_): δ 21.3, 30.6, 34.3, 39.8, 125.3, 128.5, 128.9, 134.7, 137.2, 139.4.

*2-(bromomethyl)-1-((4-isopropylphenyl)sulfonyl)aziridine (*6h*).* Colorless oil, Yield 75%. IR (KBr) (ν*_max_*/cm^−1^): 1322, 1160, 781, 696. ^1^H NMR (400.1 MHz, CDCl_3_): δ 1.27 (d, *J* = 6.8 Hz, 6H), 2.23 (d, *J* = 4.4 Hz, 1H), 2.80 (d, *J* = 6.8 Hz, 1H), 2.96–3.01 (m, 1H), 3.08–3.11 (m, 1H), 3.28 (d, *J* = 6.8 Hz, 2H), 7.40 (d, *J* = 8.0 Hz, 2H), 7.87 (d, *J* = 8.4 Hz, 2H). ^13^C NMR (100.6 MHz, CDCl_3_): δ 22.5, 29.6, 33.2, 33.3, 38.7, 126.2, 127.3, 133.6, 154.6.

*(1S,4S/R)-1-(((2-(bromomethyl)aziridin-1-yl)sulfonyl)methyl)-7,7-dimethylbicyclo[2.2.1]heptan-2-one (*6i*).* Colorless oil, Yield 81%. IR (KBr) (ν*_max_*/cm^−1^): 1742, 1325, 1146, 930, 793, 689. ^1^H NMR (400.1 MHz, CDCl_3_): δ 0.89 (s, 3H), 1.12 (s, 3H), 1.21 (t, *J* = 7.2 Hz, 1H), 1.45–1.48 (m, 1H), 1.74–1.79 (m, 1H), 2.06–2.07 (m, 1H), 2.12-2.14 (m, 1H), 2.30–2.32 (m, 1H), 2.32–2.53 (m, 2H), 2.79 (d, *J* = 6.8 Hz, 1H), 3.08–3.22 (m, 1H), 3.36–3.49 (m, 2H), 3.66–3.76 (m, 2H). ^13^C NMR (100.6 MHz, CDCl_3_): δ 15.1, 18.2, 19.61, 19.67, 19.7, 24.7, 24.8, 26.8, 31.2, 31.3, 33.6, 33.7, 39.6, 39.8, 42.4, 42.6, 48.01, 48.08, 49.0, 49.1, 57.9, 58.1, 58.2, 65.7, 214.23, 214.26.

### 4.3. Computational Method

All molecules investigated were initially subjected to a complete conformational analysis applying the Drieding forcefield within Materials Studio 2016. The 10 lowest energy conformers were optimized with VAMP (AM1) and the lowest energy structure subsequently used for DFT and TDDFT calculations within Gaussian16. All structures, unless specified otherwise, were optimized with B3LYP/6-31G(d) and confirmed minima by the absence of imaginary frequencies [25]. The MOs, NTOs and vertical excitations (TD-DFT) were calculated with additional polarization/diffuse functions (B3LYP/6-31+G(2d,p)) and with a range corrected exchange functional (CAM-B3LYP/6-31+G(2d,p)) for better treatment of potential charge transfer states [26,27,28,29]. Only NTOs with an eigenvalue (λ_i_) greater than 0.10 were analyzed.

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
