# Peer review of "Synthesis and Density Functional Theory Studies of Azirinyl and Oxiranyl Functionalized Isoindigo and (3*Z*,3’*Z*)-3,3’-(ethane-1,2-diylidene)bis(indolin-2-one) Derivatives"

_molecules, 2019, doi:10.3390/molecules24203649_

Round 1

Reviewer 1 Report

This manuscript can be accepted for publication, however the following points must be addressed: i.) the experimental details of the synthesis of 3 should be omitted from the beginning of Section 2.1.; ii.) after line 371 a brief procedure should be inserted for the preparation of reagents type 6 of which designation needs correction and supplemented with 6d as the description of this compound can not be found in the characterisation part.

Author Response

i) Although we feel that the 3 lines dedicated to the outline of the procedure for the synthesis of 3 is useful to the reader and helps with the flow, it has been removed at the request of the reviewer.

ii) The general procedure for the synthesis of 6 has been included where suggested by the reviewer. Two additional references were also added (41 and 42) to the data for compounds 6d, 6g and 6e (the spectroscopic data for 6d was mentioned by the reviewer and this is now referred to a reference).

Reviewer 2 Report

The manuscript "Synthesis and density functional theory studies of azirinyl and oxiranyl functionalized isoindigo and (3Z,3'Z)-3,3'-(ethane-1,2-diylidene)bis(indolin-2-one) derivatives" by Khalili et al reports the design and synthesis of functionalized isoindigo compounds besides employing DFT and td-DFT calculations in order for studying the impact of N-substituent on changing electronic configuration of synthesised dye.

Introduction is too succinct and should be improved by referring to the reported experimental and theoretical work. At least one paragraph should be included stating why B3LYP/6-31+G** was selected as a model for studting synthesised dyes and why NTOs was also used (for example; Why not NBO).

Results and discussion are well written reaching a good conclusion of the conducted studies. One typo is noticed at line 407 and 408 "B3LYP/6-31+G***" which should be replaced with "B3LYP/6-31+G**"

Author Response

An additional 2 paragraphs have been added to the introduction providing a more in-depth background to computational studies in the area. The second paragraph outlines the selection of model, including the use of NTO, as requested by the reviewer. This required additional references to be included, and this has changed the reference numbering throughout the entire article.

The typo at line 407 and 408 (now line 433 and 434) "B3LYP/6-31+G***" has been replaced with "B3LYP/6-31+G(2d,p)", which is a more accurate description.

Reviewer 3 Report

This manuscript presents the design and synthesis of several isoindigo compounds, which can be of interest for organic electronic materials. The electronic properties of isoindigo-thiophene molecules with various substituents have been studied theoretically,
using the Density Functional Theory (DFT) and time-dependent DFT, with accurate exchange-correlation functionals. It was found that the functional group influences various properties.

I consider this manuscript interesting, well-written, and I recommend its publication.

Author Response

No changes requested by the reviewer.